# Mediating Effects of Trait Anxiety and State Anxiety on the Effects of Physical Activity on Depressive Symptoms

**DOI:** 10.3390/ijerph20075319

**Published:** 2023-03-30

**Authors:** Masayuki Kikkawa, Akiyoshi Shimura, Kazuki Nakajima, Chihiro Morishita, Mina Honyashiki, Yu Tamada, Shinji Higashi, Masahiko Ichiki, Takeshi Inoue, Jiro Masuya

**Affiliations:** 1Department of Psychiatry, Tokyo Medical University, Shinjuku-ku, Tokyo 160-0023, Japan; 2Department of Psychiatry, Gakuji-kai Kimura Hospital, Chuo-ku, Chiba 260-0004, Japan; 3Department of Psychiatry, Tokyo Medical University Hachioji Medical Center, Hachioji-shi 193-0998, Tokyo, Japan; 4Department of Psychiatry, Ibaraki Medical Center, Tokyo Medical University, Ami-machi, Inashiki-gun, Ibaraki 300-0395, Japan

**Keywords:** physical activity, trait anxiety, state anxiety, depressive symptoms, path analysis

## Abstract

Background: Previous studies have reported that physical activity can prevent the onset of depression and reduces anxiety. In the present study, the hypothesis that total physical activity time influences depressive symptoms via state and trait anxiety was tested by a path analysis. Methods: Self-administered questionnaires were used to survey 526 general adult volunteers from April 2017 to April 2018. Demographic information, physical activity, and state and trait anxiety were investigated. Results: The association between physical activity time and depressive symptoms was expressed as a U-shape curve. The results of the covariance structure analysis showed that differences from the optimal physical activity time (DOT) had direct positive effects on state and trait anxiety. DOT affected depressive symptoms only via trait anxiety, and this was a complete mediation model. Conclusion: The present study suggests that an optimal physical activity time exists for depressive symptoms. The path model demonstrated an association between the three factors of optimal physical activity time, trait anxiety, and depressive symptoms, and the effect was fully mediated by trait anxiety.

## 1. Introduction

Previous surveys have shown a clear association between mental health and physical activity levels [1,2]. A community epidemiological study reported that regular exercise habits are associated with milder depressive symptoms [3]. Furthermore, it was suggested that higher levels of physical activity significantly prevent the onset of depression [4,5]. Several meta-analyses also reported that exercise therapy interventions for patients with depression, including older patients, are effective in improving depressive symptoms [6,7,8,9]. Furthermore, exercise that is appropriate for each individual’s ability has been shown to reduce the severity of depression [10]. The results of these studies suggest that there are certain associations between habitual physical activity and the therapeutic and preventive effects on depression, but the exact mechanism has not been elucidated to date.

Anxiety is divided into state anxiety and trait anxiety [11]. State anxiety is a transient situational response to anxiety-provoking events and changes from moment to moment. State anxiety is low when there is no or little danger, while trait anxiety is a relatively stable tendency to respond to anxiety-provoking experiences [12]. As such, trait anxiety is a relatively stable feature, which demonstrates relatively consistent individual differences in anxiety tendency and can be considered a personality trait [12]. A community epidemiological study reported that regular exercise habits are associated with milder anxiety symptoms [3]. The U.S. Department of Health and Human Services Report 2018 [13] reported strong-grade evidence that exercise reduces both state and trait anxiety. Although exercise is also effective in the treatment of anxiety and stress-associated disorders [14,15], several meta-analyses of randomized controlled trials on the effects of exercise on anxiety in subjects without psychiatric disorders have been published [16,17,18,19]. One study demonstrated that exercising for several to dozens of weeks for 30 min or more daily reduced state and trait anxiety [17]. The effect of exercise on reductions in state anxiety is even found with acute exercise [16]. Surprisingly, a meta-analysis reported that trait anxiety, which is a relatively stable measure of anxiety tendency, improved with long-term exercise [20]. In this relatively old meta-analysis, a 16-week exercise program was most effective in improving trait anxiety, whereas a shorter exercise program of 4 to 6 weeks was not [20]. A meta-analysis including more recent randomized controlled trials (RCTs) also found that exercise programs of several to a dozen weeks improved not only state anxiety but also trait anxiety [21]. In addition, a recent small RCT reported that 8 weeks of exercise treatment following WHO and American College of Sports Medicine guidelines improved trait anxiety [22]. Although the effects of several tens of minutes of daily exercise for several weeks on state and trait anxiety have been established, as indicated by these studies, whether the effect of physical activity/exercise on anxiety influences depressive symptoms, along with its dose-response effects on anxiety and depression, have not been determined to date.

Trait anxiety, which is a personality characteristic, is considered to be a risk factor for depression, and high trait anxiety influences the development of depression as well as anxiety disorders [23]. In addition to genetic factors, the hypothalamic-pituitary-adrenal system, mitochondrial function, neurotransmitters, and childhood experiences are thought to play a role in trait anxiety development, with stressful life events triggering anxiety and depression [23,24]. It has been observed in the general adult population that individuals who have experienced childhood abuse tend to have heightened levels of trait anxiety. This increased trait anxiety, in turn, exacerbates depressive rumination and the perception of negative life events, thus exacerbating the likelihood and severity of depression. Therefore, trait anxiety acts as a mediating factor in the development of depressive symptoms [25,26]. Furthermore, anxiety disorders and depression coexist at high rates in patients, with anxiety preceding depression in many cases [27,28]. Thus, various epidemiological and clinical studies have clarified a pathway by which anxiety disorders and trait anxiety precede depression and depressive states. However, the mechanism of how physical activity affects this pathway has not been clarified to date.

When considering the mechanism by which physical activity exerts its effects on reducing anxiety and depression, it is essential to investigate the impact of physical activity patterns on the intricate association between trait anxiety and depressive symptoms, as previously mentioned [23,25,26]. Although physical activity is known to be effective for reducing both trait anxiety—a relatively stable personality trait—and state anxiety, a temporary psychological symptom, knowing whether trait anxiety and state anxiety are associated with lowering depressive symptoms will elucidate the psychological mechanism by which physical activity improves depressive symptoms. These ideas indicate the possibility that either trait anxiety or state anxiety may mediate the effects of physical activity habits on depressive symptoms. If trait anxiety acts as a mediating factor, physical activity may influence personality traits, and if state anxiety acts as a mediator, physical activity may influence psychological symptom severity. However, it remains unclear as to how trait anxiety or state anxiety modulates the impact of physical activity on depressive symptoms. Additionally, there is a lack of knowledge regarding the dose-response association between physical activity and its effects on anxiety and depressive symptoms [13]. However, the existence of an optimal physical activity time and intensity for mental health was suggested in recent studies [29,30]. While insufficient physical activity has been shown to negatively impact mental health, as aforementioned, excessive physical activity may also be harmful to mental health [31] because of the stimulation of cytokines [32] and cortisol systems [33], known as an overtraining syndrome. Therefore, we focused on the mediating effects of state and trait anxiety and hypothesized that physical activity time would affect depressive symptoms via state and trait anxiety, and that an optimal physical activity time exists in this effect. To test this hypothesis, we conducted a cross-sectional study using self-administered questionnaires in adult volunteers and analyzed the data by a path analysis.

## 2. Subjects and Methods

### 2.1. Subjects

A self-administered survey method was used to gather data from adult volunteers between April 2017 and April 2018. Questionnaires and the details of this study were distributed to 1237 adults by convenience sampling through our acquaintances at Tokyo Medical University. A total of 526 (42.5%) adult volunteers participated in this study. Written consent was obtained from participants prior to their inclusion in the study. Participants were asked to provide their demographic information (age, sex, education, marital status, employment status, subjective social status, and past and current psychiatric diseases) anonymously, as well as to complete 3 questionnaires. The inclusion criteria for this study were individuals 20 years of age or older. Those who had any severe physical illness or organic brain illness were excluded from the study. Participants were informed that the study was voluntary and that there would be no negative consequences for declining to participate. Additionally, they were assured that any personal information collected would be kept anonymous to protect their identity. The study received approval from the Medical Ethics Review Committee of Tokyo Medical University (study approval number: SH3502) and was conducted in accordance with the Declaration of Helsinki (amended in Fortaleza in 2013).

### 2.2. Questionnaires

#### 2.2.1. Patient Health Questionnaire-9 (PHQ-9)

The PHQ-9, a self-administered questionnaire, was used to evaluate depression [34]. Depressive symptoms in the previous 2 weeks were rated on the following 4-point scale: “not at all (0)”, “several days (1)”, “more than half the days (2)”, and “nearly every day (3)”. The total score of the 9 items (min: 0 points—max: 27 points) was used for the analysis of depressive symptom severity. A higher score indicates a higher severity of depressive symptoms. In the present study, Cronbach’s α coefficient calculated for the total score of this scale was 0.854, indicating very high internal consistency. The Japanese version, translated and validated by Muramatsu et al., was used [35].

#### 2.2.2. State-Trait Anxiety Inventory Form Y (STAI-Y)

The STAI-Y is a self-administered questionnaire that assesses trait and state anxiety. State anxiety indicates the degree of anxiety currently being felt, whereas trait anxiety indicates a tendency toward long-term anxiety. Each of the 20 questions was rated on the following 4-point scale: “almost never (1)”, “sometimes (2)”, “often (3)”, and “almost always (4)” [11]. The total scores of trait and state anxiety (min: 20 points—max: 80 points) were used for the analysis. In the present study, Cronbach’s α coefficients calculated for the total score of the trait and state anxiety subscales were 0.925 and 0.910, respectively, indicating very high internal consistency. The Japanese version, translated and validated by Hidano et al., was used [12].

#### 2.2.3. International Physical Activity Questionnaire (IPAQ)

A short form of the IPAQ, a self-administered questionnaire [36,37], was used to assess physical activity. Subjects self-rated how many days per week and how much time per day they were active in each of the following three categories: strong physical activity, moderate physical activity, and walking continuously for at least 10 min. As existing research suggests that neither under- nor over-physical activity is desirable for mental health and that there is an optimal physical activity time [29,30,38], the weekly total physical activity time (hours) and PHQ-9 values obtained here were used to perform a regression analysis in a quadratic equation model. This regression analysis was used to obtain the optimal physical activity time (hours per week), and the difference from the optimal physical activity time (DOT) was used as an indicator for statistical analysis.

### 2.3. Statistical Analysis

Model: The present study focused on the mediating effects of state and trait anxiety and created a path model with the hypothesis that the DOT affects depressive symptoms via effects on state and trait anxiety.

Demographic data and questionnaire data were analyzed using various statistical techniques, including *t*-tests, Pearson correlation coefficient, and multiple regression analysis, with the assistance of SPSS 28 software (IBM, Armonk, NY, USA). For a multiple regression analysis, the dependent variable was a total PHQ-9 score, which was a continuous variable. We did not use the cut-off value of the PHQ-9 in this study. The independent variables were continuous or dummy variables. Dummy variables were used for the yes/no variable or sex.

A path model was built and subsequently analyzed using path analysis (covariance structure analysis with robust maximum likelihood estimation) using Mplus 8.5 software (Muthén & Muthén, Los Angeles, CA, USA). All coefficients obtained from the covariance structure analysis were standardized. Goodness-of-fit indices were not considered, as the model was saturated. Results were considered statistically significant if the *p*-value was less than 0.05.

## 3. Results

### 3.1. Associations of Demographic Characteristics and Questionnaire Scores with PHQ-9 Score in the Study Population

Table 1 shows the demographic information and differences from optimal physical activity time (DOT), STAI-Y state anxiety and trait anxiety, and PHQ-9 scores of the 526 subjects (228 men and 298 women, mean age: 41.1 ± 11.8 years). Most of them (97.7%) were employed. Table 1 shows the results of analysis of the association between demographic information, data from each questionnaire (PHQ-9 and STAI-Y), and DOT in 526 adult volunteers. The association between total weekly physical activity time and PHQ-9 score was not statistically significant by the linear equation model but was approximated by a quadratic equation regression model significantly (Figure 1). According to the regression equation, the PHQ-9 score was lowest at 25.7 total weekly physical activity time, and the PHQ-9 score was higher at both higher and lower physical activity times. Therefore, the difference from optimal physical activity time (25.7 h/week) was defined as DOT and used for further analysis.

Sex, marital status, years of education, subjective social status, history of psychiatric diseases, current psychiatric diseases, STAI-Y score of state anxiety, STAI-Y score of trait anxiety, and DOT were significantly associated with the PHQ-9 score. However, age was not significantly correlated with the PHQ-9 score.

The total weekly physical activity time with the lowest PHQ-9 was 25.7 h/week. This value was defined as the optimal physical activity time. DOT was used for the statistical analysis.

### 3.2. Multiple Regression Analysis of PHQ-9 Score

The results of multiple regression analysis with PHQ-9 score as the dependent variable are shown in Table 2. Nine independent variables were included in the analysis. The results of multiple regression analysis showed that the only significant independent variable predicting PHQ-9 score was the STAI-Y score of trait anxiety. Other independent variables, including DOT, were not significantly associated with PHQ-9 score. Using G*power (version 3.1.9.7), the post hoc power of the multiple regression analysis was 1.0.

### 3.3. Path Model Analysis of Physical Activity Time, PHQ-9 Score, and State and Trait Anxiety as Moderators

In the model, the differences from optimal physical activity time (DOT), state anxiety (STAI-Y), trait anxiety (STAI-Y), and depressive symptoms (PHQ-9) were used as observed variables and analyzed by the path model with robust maximum likelihood estimation, as shown in Figure 2.

Regarding direct effects, the DOT showed positive direct effects on both state anxiety and trait anxiety. However, the direct effect of DOT on PHQ-9 score was not significant. Trait anxiety was significantly associated with PHQ-9 score, but the association between state anxiety and PHQ-9 score was not significant.

Regarding indirect effects, the indirect effect of DOT on the PHQ-9 score via state anxiety was not significant (standardized coefficient: 0.013, *p* = 0.125), but the indirect effect via trait anxiety was significant (standardized coefficient: 0.084, *p* < 0.01). In other words, DOT affected depressive symptoms via only its effect on trait anxiety, suggesting that this mediating effect was completely mediated because the direct effect of DOT on PHQ-9 score was not significant.

The coefficient of determination for the PHQ-9 score in this model was 0.417; i.e., this model explains 41.7% of the variance in depressive symptoms.

### 3.4. Sensitivity Analyses: The Optimal Physical Activity Time of Each Age Group and the Path Model Analyses

As a sensitivity analysis, we examined the optimal physical activity time and mediation effects for two age groups: older adults (age ≥ 60) and a relatively younger group (age < 60). Appendix A displays the results of quadratic modeling for the relationship between physical activity time and PHQ-9 score in each age group. The analysis revealed that the optimal physical activity time for older adults was 20.56 h/week, while that for the younger group was 26.86 h/week. Appendix A depict the path model analyses of physical activity time, PHQ-9 score, and state and trait anxiety as moderators. The results were consistent with the main analysis and more pronounced in the sensitivity analyses, especially when assessing the effect in each age group.

## 4. Discussion

Previous studies have indicated the existence of an optimal physical activity time regarding the influence of physical activity intensity on mental health [29]. There are also reports that this optimal physical activity time is associated with lower depressive symptoms [30,38]. In the present study, we hypothesized that physical activity time affects depressive symptoms via state and trait anxiety, and that an optimal physical activity time exists in this effect, and tested this hypothesis by a path analysis. Results showed that DOT affects depressive symptoms only through the mediation of trait anxiety and that this effect was a complete mediation. This study used the total weekly physical activity time (hours/week) to measure physical activity habits. Although data are not presented, our analyses showed similar results using total physical activity strength (METs*h/week) as a measure. To our knowledge, the findings of this study have not been reported previously, and this is the first report of its kind. As indicated earlier, various studies have reported the association between physical activity habits and depressive symptoms, as well as the effect of physical activity habits on trait anxiety [13]. However, the association between physical activity habits, trait anxiety, and depressive symptoms has not been clarified to date. The results of this study hence suggest a mechanism of action by which optimal physical activity habits can reduce depressive symptoms.

Previous studies have shown that exercise according to individual ability reduces the severity of depression [10] and that exercise can improve clinical depressive symptoms [6,7,8,9], as reported by the 2018 Physical Activity Guidelines Advisory Committee [13]. Previous studies have also reported that exercise training significantly decreases trait anxiety and state anxiety [13,16,17,20,21,22]. Furthermore, the mediating effect of trait anxiety on depressive symptoms was also reported in a study demonstrating the influence of childhood maltreatment on depressive symptoms via its effect on trait anxiety [25,26]. Considering these reports, the results obtained in the present study that physical activity affects depressive symptoms via trait anxiety may be sufficient to confirm our hypothesis. However, an association between these factors has never been confirmed previously, and the results of the present study make this clear. Whereas physical activity affects depressive symptoms via trait anxiety, the mediating effect of physical activity on depressive symptoms via state anxiety was not significant. The meta-analysis by Gordon et al. [21] also demonstrated the effect of physical activity on state anxiety, and optimal physical activity improved state anxiety in the present study. However, there was no mediating association between physical activity, state anxiety, and depressive symptoms in the present study, indicating the value of the path analysis conducted in this study. In other words, our results suggest that long-term physical activity habits decrease and prevent depressive symptoms not through the transient improvement of anxiety symptoms but through effects on personality characteristics.

As mentioned in the Introduction, anxiety disorders or personality traits relating to anxiety, such as trait anxiety and neuroticism, generally precede depression clinically and in the general population [23,24,25,26,27,28]. From this psychiatric and psychological theoretical standpoint, the path model from trait anxiety to depression in the present study makes sense. Furthermore, since physical activity alleviates trait anxiety and depression, as reported in previous studies [13], the location of physical activity upstream of the path model aligns with reality. This theoretical model, in which the linkage between these factors was significant in this study, is now proposed and will provide a new understanding of prevention and intervention by physical activity.

Previous studies have reported that high physical activity significantly prevents the onset of depression [4,5] and that regular physical activity habits reduce the degree of anxiety [13]. Regarding the association between physical activity and depression, it has been observed that physical activity stimulates the process of adult neurogenesis in the dentate gyrus of the hippocampus, leading to antidepressant effects. Additionally, it has been found that the antidepressant effects of running are closely associated with both neurogenesis and plasticity of the hippocampus [39]. The association between neurogenesis and anxiety and depression has also been demonstrated in the findings of increased food avoidance in a new environment after acute stress, increased behavioral despair in the forced swim test, as well as decreased sucrose preference (a measure of anhedonia) in neurogenesis-deficient mice [40]. Animal studies have shown that physical activity increases the levels of neurotransmitters, such as serotonin, noradrenaline, and dopamine [41,42,43], and it causes neural changes that may improve depressive symptoms [41]. The finding of the present study that trait anxiety acts as a complete mediator of the effect of physical activity on depressive symptoms may be attributed to the effects of physical activity on increasing neurogenesis, as suggested by the results of previous studies.

The present study demonstrated that trait anxiety acts as a mediator in the association between physical activity and depressive symptoms. Another study has also reported that rumination plays a mediating role in the association between trait anxiety and depressive symptoms [26]. Furthermore, although limited to women, it has been reported that yoga and aerobic exercise significantly improved the severity of depressive symptoms and that its effect was mediated by a decrease in rumination [44]. Other studies reported that combined meditation and aerobic exercise training reduced rumination in medical students [45] and reduced depressive symptoms and rumination [46]. The results of our present study suggest that optimal physical activity time improves depressive symptoms through the mediation of trait anxiety, and it was also reported that exercise affects depressive symptoms through its effect on rumination [44]. It was also reported that rumination mediates the association between trait anxiety and depressive symptoms [26]. Taken together, these results suggest that optimal physical activity habits may influence rumination via trait anxiety and further improve depressive symptoms. There have been several reports on the mediating effects of rumination, including reports that experiences of maltreatment in childhood worsen depressive symptoms and unpleasant moods via enhancing rumination in adulthood [26,47,48,49]. Future studies may clarify whether the improvement of rumination mediates the anti-depressive effects of optimal physical activity habits via improving trait anxiety.

The present analysis showed that the effect of optimal physical activity time on depressive symptoms was mediated by trait anxiety and that the mediating effect of trait anxiety was complete mediation. Although the association between physical activity and mental health has been reported in various studies, as described in the Introduction section, the dose-response relationship between physical activity and mental health has not been clarified, which makes specific interventions, such as exercise therapy, difficult. Whereas positive associations between physical activity and mental health have been reported, it was also reported that excessive physical activity levels exacerbate mental health burden as much as no physical activity at all [29,30]. In addition, a U-shaped association between physical activity for leisure and depressive symptoms among Japanese workers [38] was also reported. These data suggest that there is an optimal physical activity time in the association between physical activity and mental health. Therefore, it can be expected that optimal physical activity will alleviate trait anxiety, further improving depressive symptoms. The index of DOT presented in this study may be used as a concrete indicator of this “optimal physical activity” and is considered highly important clinically.

This cross-sectional study needs to be validated in prospective studies to conclude a causal association. As this study was conducted on adult volunteers collected by convenience sampling from acquaintances, which may not represent the general adult population, the obtained results may not apply to patients with depression. This study was conducted for Japanese adults. However, the results of this study can be extrapolated to the population of other age strata and other countries, although the replication should be confirmed. Furthermore, the self-administered questionnaires are subjective evaluations, and the results may diverge from objective assessments. In particular, as physical activity evaluation in this study was not based on an objective evaluation, there are limitations to the conclusions. Future studies using objective evaluation methods for physical activity are needed. In the present study, optimal physical activity time was identified for the minimalize depressive symptoms, but an optimal physical activity time also exists for the minimalize trait anxiety (data not shown). As U-shaped quadratic equation models represent both, physical activity times that are both too long and too short will not have favorable effects on depressive symptoms and trait anxiety. However, as the optimal physical activity time for each is slightly different, it may be necessary to tune the optimal time for both. The Physical Activity Guidelines Advisory Committee 2018 also stated that the linearity between physical activity, trait anxiety, and depression has not been verified [13]. Finally, although we have assessed all kinds of physical activity, it is important to note that the relationships of the dose-response curve between mental health and intentional physical activity, namely, “exercise”, may differ. Therefore, caution should be exercised when adopting the results to intervention programs.

## 5. Conclusions

This study suggests that trait anxiety fully mediates the influence of optimal physical activity habits on depressive symptoms. Furthermore, the path model demonstrated a tripartite association between physical activity, trait anxiety, and depressive symptoms.

## Figures and Tables

**Figure 1 ijerph-20-05319-f001:**
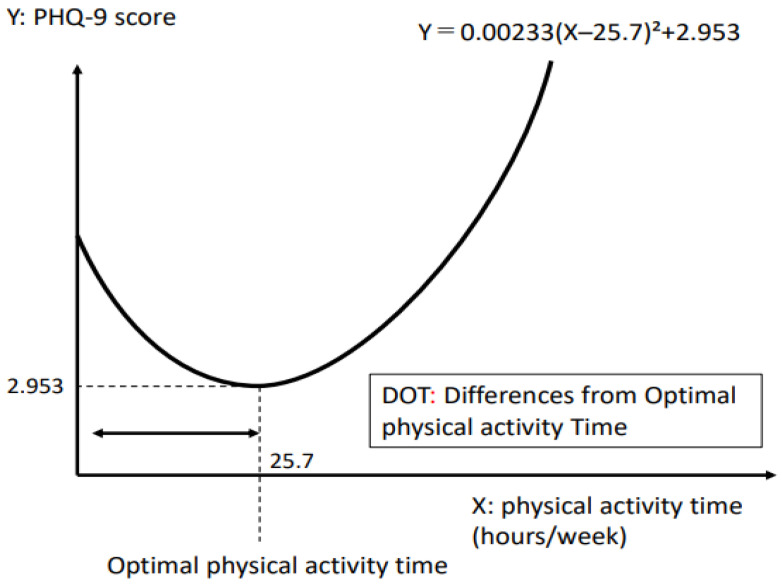
Correlation between PHQ-9 score and physical activity time.

**Figure 2 ijerph-20-05319-f002:**
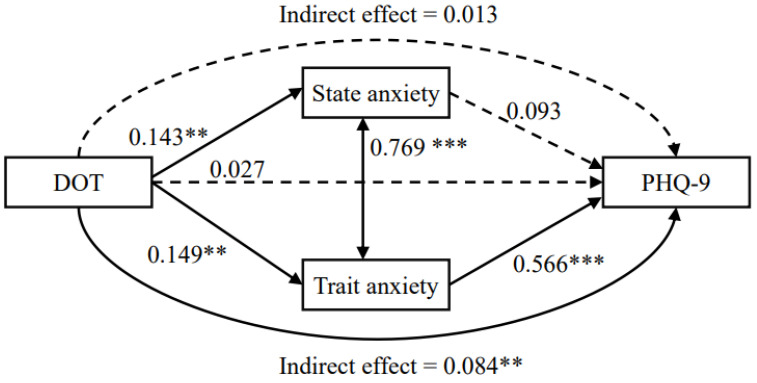
Results of the path model using DOT (differences from optimal physical activity time), STAI-Y (state and trait anxiety), and PHQ-9 (depressive symptom severity). Direct and indirect effects between the variables are shown. Values indicate standardized coefficients. ** *p* < 0.01, *** *p* < 0.001.

**Table 1 ijerph-20-05319-t001:** Association of demographic information and questionnaire data with the severity of depressive symptoms (PHQ-9 score) in 526 adult volunteers.

Characteristic or Measure	Value (Number or Mean ± SD)	Correlation with PHQ-9 (*r*) or Effect on PHQ-9
Age	41.1 ± 11.8 (Range: 20–77; 20–29, *n* = 100; 30–39, *n* = 154; 40–49, *n* = 136; 50–59, *n* = 90; 60-, *n* = 45)	*r* = –0.028, *p* = 0.265
Sex (men:women)	228:298	Men 3.3 ± 3.8 vs. women 4.5 ± 4.4, *p* ˂ 0.001 (*t*-test)
Married (yes:no)	346:176	Yes 3.5 ± 3.9 vs. no 5.0 ± 4.4, *p* < 0.001 (*t*-test)
Education years	14.7 ± 1.8	*r* = –0.093, *p* = 0.018
Subjective social status (lowest: 1 to highest: 10)	5.1 ± 1.7	*r* = –0.267, *p* < 0.001
Past history of psychiatric diseases (yes:no)	62:464	Yes 6.7 ± 5.4 vs. no 3.6 ± 3.8, *p* < 0.001 (*t*-test)
Current psychiatric disease (yes:no)	21:496	Yes 8.1 ± 5.0 vs. no 3.8 ± 4.1, *p* < 0.001 (*t*-test)
STAI-Y score (state anxiety)	41.2 ± 9.7	*r* = 0.548, *p* < 0.001
STAI-Y score (trait anxiety)	43.0 ± 10.5	*r* = 0.645, *p* < 0.001
DOT (hours/week)	19.5 ± 7.5	*r* = 0.124, *p* = 0.004
PHQ-9 score	4.0 ± 4.2	

Data are presented as means ± SD or numbers. *r* = Pearson correlation coefficient. PHQ-9: Patient Health Questionnaire-9; STAI-Y: State-Trait Anxiety Inventory form Y; DOT: Differences from Optimal physical activity Time.

**Table 2 ijerph-20-05319-t002:** Multiple regression analysis of the factors associated with depressive symptoms (PHQ-9 score).

Independent Variable	Beta	*p*-Value	VIF
STAI-Y score (trait anxiety)	0.515	<0.001	2.796
STAI-Y score (state anxiety)	0.096	0.076	2.618
Past history of psychiatric disease (yes 2 vs. no 1)	0.084	0.031	1.367
Current psychiatric disease (yes 2 vs. no 1)	0.078	0.042	1.316
Sex (1 male, 2 females)	0.073	0.037	1.115
Subjective social status	–0.057	0.139	1.323
Education years	0.034	0.405	1.514
Age	–0.005	0.901	1.282
DOT	0.003	0.939	1.069

Adjusted *R*^2^ = 0.437, *F* = 44.745, *p* < 0.001. Beta = standardized partial regression coefficient. VIF, variance inflation factor; PHQ-9, Patient Health Questionnaire-9; STAI-Y, State-Trait Anxiety Inventory form Y; DOT, Differences from Optimal physical activity Time. Independent factors: STAI-Y score (state anxiety), STAI-Y score (trait anxiety), history of psychiatric disease, current psychiatric disease, sex, subjective social status (lowest: 1 to highest: 10), education years, age, and DOT.

## Data Availability

The data presented in this study are available on request from the corresponding author. The data are not publicly available due to this study belongs to an ongoing research project.

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
