# Peer review of "Mediating Effects of Trait Anxiety and State Anxiety on the Effects of Physical Activity on Depressive Symptoms"

_ijerph, 2023, doi:10.3390/ijerph20075319_

Round 1
Reviewer 1 Report
Dear authors, the article tackles an important issue, please find my comments
· “On the other hand, trait anxiety 45 is the tendency to perceive various threatening situations in the same way and to react to such situations similarly. Please rephrase as the sentence is unclear
· The settings of the study are unclear, from where the participants were collected?
· How many participants were approached? how many declined? Etc…
· The results section does not contain a section or a table to describe the respondents’ characteristics
· “Multiple regression analysis with PHQ-9 score as the dependent vari-188 able are shown in Table 2. Nine independent variables were included in the analysis” for me it is unclear, is it a linear regression? Or a binary logistic? Was the PHQ-9 score treated as a continuous variable? Not cut-off values were used, also in regards to the independent variables they are categorical? Please clarify
· Are the results applicable only for Japanese Adults? How are the findings related at global level?
Author Response
“On the other hand, trait anxiety is the tendency to perceive various threatening situations in the same way and to react to such situations similarly. Please rephrase as the sentence is unclear.
Response: Thank you for the comment. As indicated, we defined “trait anxiety” more clearly as follows .
“On the other hand, trait anxiety is a relatively stable tendency to respond to anxiety-provoking experiences.”(Ref12 Hidano)
The settings of the study are unclear, from where the participants were collected? How many participants were approached? how many declined? Etc…
Response: As indicated, we added from where the participants were collected and how many participants were approached and declined as follows.
“Questionnaires and the details of this study were distributed to 1237 adults by convenience sampling through our acquaintances at Tokyo Medical University. A total of 526 (42.5%) adult volunteers participated in this study. Written consent was obtained from participants prior to their inclusion in the study.”
The results section does not contain a section or a table to describe the respondents’ characteristics.
Response: As indicated, we added the description of the respondents’ characteristics as follows.
“Table 1 shows the demographic information, and differences from optimal physical activity time (DOT), STAI-Y state anxiety and trait anxiety, and PHQ-9 scores of the 526 subjects (228 men and 298 women, mean age: 41.1 ± 11.8 years). Most of them (97.7%) were employed.”
“Multiple regression analysis with PHQ-9 score as the dependent variable are shown in Table 2. Nine independent variables were included in the analysis” for me it is unclear, is it a linear regression? Or a binary logistic? Was the PHQ-9 score treated as a continuous variable? Not cut-off values were used, also in regards to the independent variables they are categorical? Please clarify
Response: We conducted a multiple linear regression analysis: the dependent variable is a total PHQ-9 score, which is a continuous variable. We did not use the cut-off value of the PHQ-9 in this study. The independent variables were continuous or dummy variables. Dummy variables were used for the yes/no variable or sex. We added the following descriptions to the Methods section and Table 2.
“For a multiple regression analysis, the dependent variable was a total PHQ-9 score, which was a continuous variable. We did not use the cut-off value of the PHQ-9 in this study. The independent variables were continuous or dummy variables. Dummy variables were used for the yes/no variable or sex.”
“yes 2, no 1” in Table 2
Are the results applicable only for Japanese Adults? How are the findings related at global level?
Response: This study was conducted for Japanese adults. However, the results of this study can be extrapolated to the population of other age strata and other countries, although the replication should be confirmed. The following description was added to the Discussion section as a limitation of the study.
“This study was conducted for Japanese adults. However, the results of this study can be extrapolated to the population of other age strata and other countries, although the replication should be confirmed.”
Reviewer 2 Report
Dear Authors,
The topic is potentially important. Thanks for this well written work.
Abstract: Sufficient
Introduction: Sufficient
Material and Methods:
Line 113-114: Post-hoc power can be calculated.
Line 125: More information can be given about the scales.
Discussion: Sufficient
Author Response
Line 113-114: Post-hoc power can be calculated.
Response: Using G*power (version 3.1.9.7), post-hoc power of the multiple regression analysis was 1.0. The following description was added to the Results section.
“Using G*power (version 3.1.9.7), post-hoc power of the multiple regression analysis was 1.0.”
Line 125: More information can be given about the scales.
Response: As indicated, more information was added to the scales as follows.
“The total score of the 9 items (min: 0 points – max: 27 points) was used for the analysis of depressive symptom severity. A higher score indicates a higher severity of depressive symptoms. In the present study, Cronbach’s α coefficient calculated for the total score of this scale was 0.854, indicating very high internal consistency.”(PHQ)
“The total scores of trait and state anxiety (min: 20 points – max: 80 points) were used for the analysis. In the present study, Cronbach’s α coefficients calculated for the total score of the trait and state anxiety subscales were 0.925 and 0.910, respectively, indicating very high internal consistency.”(STAI-Y)
Reviewer 3 Report
Dear Author
The title of the paper is interesting and novel. I have enjoyed reading this research paper exploring how moderate physical activity effects depressive symptoms via anxiety. The topic is novel. The abstract can be rewritten as the flow is missing.
There are few suggestions which may help to improve the manuscript. Most of the suggestions have been given in file however some over all observations are mentioned below in pointers
1. author is using three different terms for physical activity- physical activity time, exercise and physical activity habit. It would be better if author can clearly state the difference and present a definition of this variable. as it is the most important variable in the current study
2. while presenting the rational and theoretical framework author tried to connect it with two three different ways- one is through studies done on exercise, leisure, physical activity and biological (brain pathways). Author can propose a clear theoretical stand to rationalize the study
3. some more clarity on demographic data which has been inserted in to comment box in article can improve the results
4. age group can itself be a great determinant of physical activity and inclusion of such large number of age group raises serious questions on outcomes therefore a categorical analysis can be a good idea to avoid this
5. in discussion a clear theoretical stance and rationale is missing
6. overall article is written well however a better flow is needed to report the findings well.

Author Response
1. author is using three different terms for physical activity- physical activity time, exercise and physical activity habit. It would be better if author can clearly state the difference and present a definition of this variable. as it is the most important variable in the current study
Response: Thank you for pointing that out. For the meaning of "exercise" referring to intentional physical activity, we will continue to use the word "exercise". As for the vague expression "physical activity", we have unified the wording.
We also added to the following descriptions to show our theoretical standpoint to rationalize our study more clearly in the Discussion.
“Finally, although we have assessed all kinds of physical activity, it is important to note that the relationships of the dose-response curve between mental health and intentional physical activity, namely, "exercise", may differ. Therefore, caution should be exercised when adopting the results to intervention programs.”
2. while presenting the rational and theoretical framework author tried to connect it with two three different ways- one is through studies done on exercise, leisure, physical activity and biological (brain pathways). Author can propose a clear theoretical stand to rationalize the study
Response: Thank you for your advice. We added to the following descriptions to show our theoretical standpoint to rationalize our study more clearly in the Discussion.
“As mentioned in the Introduction, anxiety disorders or personality traits relating to anxiety, such as trait anxiety and neuroticism, generally precede depression clinically and in the general population [23-28]. From this psychiatric and psychological theoretical standpoint, the path model from trait anxiety to depression in the present study makes sense. Furthermore, since physical activity alleviates trait anxiety and depression, as reported in previous studies [13], the location of physical activity upstream of the path model aligns with reality. This theoretical model, in which the linkage between these factors was significant in this study, is now proposed and will provide a new understanding of prevention and intervention by physical activity.”
3. some more clarity on demographic data which has been inserted in to comment box in article can improve the results
Response: As indicated, the following description was added to the Methods section and Table 1.
“Participants were asked to provide their demographic information (age, sex, education, marital status, employment status, subjective social status, past and current psychiatric diseases) anonymously as well as 3 questionnaires to complete.” (in the Methods section)
Age (range 20–77; 20-29, n=100; 30-39, n=154; 40-49, n=136; 50-59, n=90; 60-, n=45) in Table 1.
4. age group can itself be a great determinant of physical activity and inclusion of such large number of age group raises serious questions on outcomes therefore a categorical analysis can be a good idea to avoid this
Response: We agree the reviewer's concern and we have added the sensitivity analyses divided by age group (Older adult: age>=60, and relatively younger group: <60). As shown in the supplementary table 1, optimal physical activity time of older adult group (20.56 hours/week) is different from that of younger group (26.86 hours/week). The mediation effects were also examined. As shown in supplementary figure 1 and 2, the result is consistent and more emphasized in the sensitivity analyses.
“3.4. Sensitivity analyses: the optimal physical activity time of each age group and the path model analyses.
As a sensitivity analysis, we examined the optimal physical activity time and media-tion effects for two age groups: older adults (age ≥ 60) and a relatively younger group (age < 60). Supplementary Table 1 displays the results of quadratic modeling for the relation-ship between physical activity time and PHQ-9 score in each age group. The analysis re-vealed that the optimal physical activity time for older adults was 20.56 hours/week, while that for the younger group was 26.86 hours/week. Supplementary Figures 1 and 2 depict the path model analyses of physical activity time, PHQ-9 score, and state and trait anxiety as moderators. The results were consistent with the main analysis and more pro-nounced in the sensitivity analyses, especially when assessing the effect in each age group.”
Supplementary table 1 and supplementary figure 1 and 2: added.
5. in discussion a clear theoretical stance and rationale is missing
Response: Thank you for your suggestion. The following description, which was already reported in our response to your comment #2, was added to the Discussion section.
“As mentioned in the Introduction, anxiety disorders or personality traits relating to anxiety, such as trait anxiety and neuroticism, generally precede depression clinically and in the general population [23-28]. From this psychiatric and psychological theoretical standpoint, the path model from trait anxiety to depression in the present study makes sense. Furthermore, since physical activity alleviates trait anxiety and depression, as reported in previous studies [13], the location of physical activity upstream of the path model aligns with reality. This theoretical model, in which the linkage between these factors was significant in this study, is now proposed and will provide a new understanding of prevention and intervention by physical activity.”
6. overall article is written well however a better flow is needed to report the findings well.
Response: Thank you for your advice. We have taken the reviewer's feedback into consideration, and as a result, we believe that the logical flow of the paper has become more coherent.